# The Cannabinoid-Like Compound, VSN16R, Acts on Large Conductance, Ca^2+^-Activated K^+^ Channels to Modulate Hippocampal CA1 Pyramidal Neuron Firing

**DOI:** 10.3390/ph12030104

**Published:** 2019-07-04

**Authors:** Setareh Tabatabaee, David Baker, David L. Selwood, Benjamin J. Whalley, Gary J. Stephens

**Affiliations:** 1Reading School of Pharmacy, University of Reading, Reading RG6 6AP, UK; 2Centre for Neuroscience and Trauma, Blizard Institute, Queen Mary University of London, London E1 4AT, UK; 3Department of Medicinal Chemistry, UCL Wolfson Institute for Biomedical Research, University College London, London WC1E 6BT, UK

**Keywords:** BK_Ca_ channels, hippocampal pyramidal neurons, VSN16R, cannabinoid, 7-pra-martentoxin

## Abstract

Large conductance, Ca^2+^-activated K^+^ (BK_Ca_) channels are widely expressed in the central nervous system, where they regulate action potential duration, firing frequency and consequential neurotransmitter release. Moreover, drug action on, mutations to, or changes in expression levels of BK_Ca_ can modulate neuronal hyperexcitability. Amongst other potential mechanisms of action, cannabinoid compounds have recently been reported to activate BK_Ca_ channels. Here, we examined the effects of the cannabinoid-like compound (R,Z)-3-(6-(dimethylamino)-6-oxohex-1-en-1-yl)-N-(1-hydroxypropan-2-yl) benzamide (VSN16R) at CA1 pyramidal neurons in hippocampal ex vivo brain slices using current clamp electrophysiology. We also investigated effects of the BK_Ca_ channel blockers iberiotoxin (IBTX) and the novel 7-pra-martentoxin (7-Pra-MarTx) on VSN16R action. VSN16R (100 μM) increased first and second fast after-hyperpolarization (fAHP) amplitude, decreased first and second inter spike interval (ISI) and shortened first action potential (AP) width under high frequency stimulation protocols in mouse hippocampal pyramidal neurons. IBTX (100 nM) decreased first fAHP amplitude, increased second ISI and broadened first and second AP width under high frequency stimulation protocols; IBTX also broadened first and second AP width under low frequency stimulation protocols. IBTX blocked effects of VSN16R on fAHP amplitude and ISI. 7-Pra-MarTx (100 nM) had no significant effects on fAHP amplitude and ISI but, unlike IBTX, shortened first and second AP width under high frequency stimulation protocols; 7-Pra-MarTx also shortened second AP width under low frequency stimulation protocols. However, in the presence of 7-Pra-MarTx, VSN16R retained some effects on AP waveform under high frequency stimulation protocols; moreover, VSN16R effects were revealed under low frequency stimulation protocols. These findings demonstrate that VSN16R has effects in native hippocampal neurons consistent with its causing an increase in initial firing frequency via activation of IBTX-sensitive BK_Ca_ channels. The differential pharmacological effects described suggest that VSN16R may differentially target BK_Ca_ channel subtypes.

## 1. Introduction

The cannabinoid-like compound (R,Z)-3-(6-(dimethylamino)-6-oxohex-1-en-1-yl)-N-(1-hydroxypropan-2-yl) benzamide (VSN16R) [1] has recently been shown to represent a potential therapeutic lead compound for spasticity [2]. VSN16R has a promising pharmacokinetic and safety profile, and can penetrate the CNS. In terms of mechanism of action, VSN16R has been shown to lack effects at cannabinoid CB_1_ and CB_2_ and at GPR55 receptors, but rather to act as an opener of large conductance, Ca^2+^-activated K^+^ (BK_Ca_) channels [2]. However, reports of VSN16R effects on BK_Ca_ channels in native CNS neurons are currently lacking.

Native BK_Ca_ channels are formed by the tetramers of the pore-forming α-subunit encoded by the *Kcnma1* gene, which can associate with auxiliary β (β1-4) and, potentially, γ (γ1-4) subunits [3,4]. Amongst voltage-gated K^+^ channels, BK_Ca_ are unique in that they are both voltage-regulated and Ca^2+^-regulated. Opening of BK_Ca_ channels is predicted to decrease neuronal excitability, and VSN16R has been proposed to counteract spasticity by such a mechanism [2]. VSN16R was originally synthesized as an analogue of anandamide [1], a predominant endocannabinoid in the mediation of endocannabinoid tone in the mammalian hippocampus [5]. However, VSN16R also has structural similarities to the endocannabinoid-like lipids, N-arachidonoyl glycine and N-arachidonoyl serine, which can both also act as BK_Ca_ openers [6], suggesting a potential alternative mechanism for cannabinoid signalling. There is also evidence of differential VSN16R effects on BK_Ca_ subunits; thus, VSN16R has been proposed not to activate KCNMA1 directly, but rather may activate channels expressing a β4 auxiliary subunit [2]. Therefore, in this short communication, we investigated whether VSN16R affected neuronal excitability in hippocampal CA1 pyramidal neurons, an area associated with aberrant activity in epilepsy, and the pharmacological sensitivity of any VSN16R effects to a standard BK_Ca_ blocker, IBTX, and the novel BK_Ca_ blocker, 7-Pra-MarTx, in these neurons.

## 2. Methods

### 2.1. Ethical Approval

All work was subject to local University of Reading Animal Welfare Ethical Review Body approval and was conducted in accordance with the UK Animals (Scientific Procedures) Act, 1986 and ARRIVE guidelines; every effort was made to minimise pain and discomfort experienced by animals.

### 2.2. Preparation of Acute Hippocampus Slices

Acute hippocampal slices (400 μm) were prepared from healthy, male, P28-P42 C57BL/6 mice (Charles River Ltd., UK) using a VT1200 Vibratome (Leica, UK). Animals were anaesthetized by inhalation of isoflurane (2%), and euthanized by cervical dislocation followed by decapitation. The brain was then rapidly removed and submerged in cold (4 °C) artificial cerebrospinal fluid (aCSF) comprising (in mM): NaCl 126, D-glucose 10, MgCl_2_ 2, KCl 2.5, NaH_2_PO_4_ 1.25, NaHCO_3_ 26, CaCl_2_ 2; pH adjusted to 7.25 with 1 M KOH. All reagents were obtained from Fisher Scientific UK Ltd. (Loughborough, UK). Slices were maintained under carboxygenated (95% O_2_/5% CO_2_) standard aCSF at 37 °C for <1 h before being returned to room temperature (24 °C). Recordings were made at 24 °C, 2–8 h following slice preparation.

### 2.3. Electrophysiology

Individual hippocampal brain slices were placed in a recording chamber maintained at room temperature (24 °C) and superfused with carboxygenated standard aCSF. CA1 region pyramidal neurons were identified anatomically and morphologically using an IR-DIC upright Olympus BX50WI microscope (Olympus, Tokyo, Japan) with a 60× numerical aperture 0.9, water immersion lens. Whole-cell patch-clamp recordings from pyramidal neurons were made in current clamp mode with an EPC-9 patch-clamp amplifier (HEKA Electronik, Lambrecht, Germany) using Patchmaster v2.43 data acquisition software (HEKA Electronik) on a Macintosh G4 computer (Apple Computer, Cupertino, CA, USA). Electrodes were fabricated from borosilicate glass (GC150-F10, Harvard Apparatus, Kent, UK) using a horizontal electrode puller (P-87 Flaming/Brown micropipette puller; Sutter Instruments Co., California, USA) and fire-polished using a micro-forge (MF-830; Narishige; Japan). Patch pipettes had tip resistances in the range of 5–7 MΩ when backfilled with an intracellular solution comprised of (in mM): KCl 150, CaCl_2_ 1, MgCl_2_ 1 (Fisher Scientific, UK), K_2_ATP 4, NaGTP 0.3, HEPES 10 (all Sigma Aldrich, UK unless stated) and pH adjusted to 7.25 with 1 M KOH. Prior to stimulation, the membrane potential was maintained at a constant level (–70 mV) by injection of DC current. Experiments were performed with two different high- and low-frequency stimulus protocols in order to evoke action potential (AP) firing: strong/short pulse (high frequency, 0.35–0.5 nA, 50 ms) and weak/long pulse (low frequency, 0.15–0.25 nA, 100 ms). Characterization of AP waveforms was performed on the first and second spike of a train (of three or more spikes) triggered by minimal current injection. The input resistance was regularly monitored using a current injection of –20 pA from –70 mV holding current. The amplitude of fAHP in terms of absolute change was measured from the membrane holding potential voltage to the most negative point after repolarization. ISIs were measured as the time between evoked AP peak to the subsequent AP peak.

### 2.4. Pharmacology

The putative BK_Ca_ channel opener VSN16R (supplied by Canbex Therapeutics, UK) and the BK_Ca_ channel blockers, iberiotoxin (IBTX) (Alomone Labs, Israel) and 7-pra martentoxin (7-Pra-MarTx, supplied by Canbex Therapeutics, UK) were used. 100 μM VSN16R was used a saturating concentration [2] to ensure action in brain slice preparation used, where potential lipophilic can reduced apparent efficacy [7]. 100 nM IBTX [8] and 100 nM 7-Pra-MarTx was used at an effective concentration of the parental martentoxin [9,10]. 7-Pra-MarTx is modified with a propargylglycine substitution at the Lys7 residue of martentoxin [9] which is predicted from the 3D structure to be distant from required binding residues and has the sequence FGLIDV[Pra]CFASSECWTACKKVTGSGQGKCQNNQCRCY, with disulfide connectivity at C8–C29, C14–C34, and C18–C36 (see Figure 3A). All drugs were made up as 1000× stocks and stored as aliquots at –20 °C prior to use. Drugs were diluted in carboxygenated standard aCSF immediately before use, and bath applied ≥20 min to reach steady state before each recording.

### 2.5. Data Analysis

For analysis, data were initially exported for analysis using Fitmaster software v2.90.1 (HEKA). Statistical analysis was performed using GraphPad Prism (v5, GraphPad Software Inc., La Jolla, CA). Data were tested for normality and shown to be normally distributed (*P* < 0.05, D’Agostino and Pearson omnibus normality test). Comparison of paired treatment groups was performed using paired two-tailed t-test, or comparison of multiple treatment groups was performed using repeated measures ANOVA followed by Newman–Keuls post hoc test.

## 3. Results

### 3.1. VSN16R Affects AP Waveform during High Frequency Stimulation in Hippocampal CA1 Pyramidal Neurons

Bath application of VSN16R (100 μM) had significant effects on AP waveforms triggered by a high frequency (0.35–0.5 nA, 50 ms) protocol in CA1 neurons (Table 1; Figure 1). By contrast, AP waveforms triggered by a low frequency (0.35–0.5 nA, 50 ms) protocol were unaffected by VSN16R (100 μM) (Table 1). VSN16R (100 μM) increased first and second fAHP amplitude stimulated by a high frequency protocol, and decreased first and second ISI stimulated by a high frequency protocol (Table 1, Figure 1). VSN16R (100 μM) also reduced first AP width. VSN16R (100 μM) also caused a significant reduction in the amplitude of the first AP (from 111.2 ± 2.7 mV to 107.4 ± 3.8 mV, n = 7, *P* < 0.05). Together, these data are consistent with VSN16R acting as a BK_Ca_ channel opener to increase channel activity in response to high frequency stimulation.

### 3.2. VSN16R Effects Are Blocked by IBTX, but Effects Persist, or Are Uncovered, in the Presence of 7-Pra-MarTx

Bath application of the BK_Ca_ channel blocker IBTX (100 nM) produced effects on AP waveforms triggered by the high frequency protocol in CA1 neurons (Table 2). Most notably, IBTX decreased first fAHP amplitude, increased second IS1 and broadened first AP width (Table 2, Figure 2). Importantly, VSN16R (100 μM) could no longer modulate AP waveforms (with the exception of an effect on first AP width) in the presence of IBTX. IBTX had no effects on fAHP and ISI parameters triggered by the low frequency protocol, but IBTX broadened first and second AP width. Similar to previous results, VSN16R (100 μM) had no further effects under the low frequency protocol in the presence of IBTX.

Bath application of the novel BK_Ca_ channel blocker 7-Pra-MarTx (Figure 3A) (100 nM) had no effect on first and second fAHP and first and second ISI parameters triggered by either the high or low frequency protocol in pyramidal neurons (Table 3, Figure 3B). However, 7-Pra-MarTx (100 nM) did cause a shortening of first and second AP width triggered by the high frequency protocol (Table 3, Figure 3B), and a shortening of second AP width triggered by the low frequency protocol (Table 3). VSN16R (100 μM) retained several actions on AP waveforms in the presence of 7-Pra-MarTx under the high frequency protocol. Thus, similarly to previous results, VSN16R (100 μM) was able to increase second fAHP amplitude, decrease first ISI and shorten first AP width stimulated by a high frequency protocol. Of further interest here was that VSN16R (100 μM) was now able to increase first and second fAHP amplitude and decrease first and second ISI stimulated by a low frequency protocol in the presence of 7-Pra-MarTx (Table 3, Figure 3C). Thus, our data suggest that some VSN16R effects under the low frequency protocol were uncovered in the presence of 7-Pra-MarTx.

## 4. Discussion

This study investigated the effects of the putative BK channel opener VSN16R on neuronal excitability of CA1 pyramidal neurons. We demonstrated for the first time in native mammalian neurons that VSN16R increased fAHP, decreased ISI and shortened AP width, effects consistent with increased BK_Ca_ activation. VSN16R actions were dependent on stimulus protocols used to induce neuronal firing, such that VSN16R effects, when applied alone, were seen at higher, but not lower, frequency stimulations. Such effects of VSN16R were blocked by IBTX, with IBTX itself having effects on the AP waveform. The latter data are consistent with previous reports of IBTX action at BK_Ca_ channel activity in CA1 neurons [8,11]. This study extends recent demonstrations that VSN16R increased single channel activity of large conductance BK_Ca_ channels in an excised inside-out patch from a human endothelial cell line or in human cerebral cortex neural HCN-2 cell lines [2] to a native neuronal preparation. This study further confirmed that cannabinoid-like compounds include BK_Ca_ channels amongst their repertoire of molecular targets.

BK_Ca_ channels are widely reported to show three major biophysical phenotypes, dependent on expression of BK_Ca_ α subunits with different accessory β subunits; thus, inactivating BK_Ca_ channels are composed of α + β2, non-inactivating type I channels of α alone and type II channels of α + β4 (reviewed in [12]). Related to this, β2 and β4 are the predominant BK_Ca_ β subunits expressed in the CNS [13], and it has been proposed that β2 expression is associated with IBTX sensitivity, while β4 expression confers IBTX resistance [14,15,16]. With further relevance here, original studies have shown that VSN16R blocked BK_Ca_ channels in Ea.Hy926 cells, which predominantly express α + β4 subunits, but had no action at BK_Ca_ channels in porcine aorta endothelium, which reportedly lack all β subunits [2]. CA1 neurons express inactivating, IBTX-sensitive BK_Ca_ channels [8,17]. The present data demonstrate that VSN16R actions in CA1 neurons are sensitive to IBTX block, in agreement with data in rat mesenteric arteries, where VSN16R induced vasorelaxation in an endothelium-dependent and IBTX-sensitive manner [2]. Together, these studies argue against a totally β4-selective activity for VSN16R. This argument is supported further by the first use of 7-Pra-MarTx, a toxin modified at the Lys7 residue of martentoxin. Martentoxin is a 37 amino acid peptide purified from scorpion (*Buthus martensi* Karsch) venom, which has been proposed as a toxin with selectivity for BK_Ca_ channels composed of α + β4, over those composed of the α subunit alone [10,18]. In the Shi et al. study [10], martentoxin inhibition was shown to be lost under conditions of increased cytoplasmic Ca^2+^. However, martentoxin also blocks BK_Ca_ in adrenal chromaffin cells [9] which may express α + β2 subunits. Conversely, martentoxin can enhance glioma BK_Ca_ currents and those composed of α + β1 subunits in an expression system [19]; martentoxin also acted in a Ca^2+-^-dependent fashion in this study, such that increased cytoplasmic Ca^2+^ promoted current potentiation. These data suggest that specific toxins, such as martentoxin and their analogues, may be useful pharmacological agents to discriminate BK_Ca_ subtypes, but that intracellular Ca^2+^ dynamics, widely known to modulate BK_Ca_ activity, can also modulate toxin effects (see also [18]). 7-Pra-MarTx, at least partially, co-localises with β4 subunits in Ea.Hy926 cells (Darryl Overby and Jacques Bertrand, Imperial College London, personal communication). Here, 7-Pra-MarTx had no major effects on fAHP or ISI, but did have some actions, causing a shortening of AP width; of note here is that 7-Pra-MarTx effects on AP width occurred in the opposite direction to those of the proposed β2-subunit-preferring IBTX. Given that CA1 neurons are reported to express β4 subunits, albeit at lower levels than the prominent β4 expression reported in other hippocampal neurons such as dentate gyrus and CA3 pyramidal neurons [20,21,22], the lack of effect of 7-Pra-MarTx on VSN16R-mediated effects is also consistent with VSN16R not being a β4 selective agent.

Of further interest here was that VSN16R had apparent addition effects in the presence of 7-Pra-MarTx. Thus, VSN16R actions under low frequency stimulation protocol were uncovered; here, VSN16R-mediated increased fAHP and decreased first ISI (as seen for VSN16R when applied alone under high, but not low, frequency stimulation). These actions are predicted to reduce neuronal hyperexcitability. A putative explanation of these data is that 7-Pra-MarTx block of β4 subunits, which are regarded as inhibitory subunits [12], further reveals VSN16R action at β2 (and/or α) channels that coexist in CA1 neurons. In this regard, different BK_Ca_ channel subtypes are reported to coexist in different neurons, including mossy fiber presynaptic terminals [23] and Purkinje cells [24], and BK_Ca_ ligand sensitivity is often associated with subpopulations of given neurons, for example, as shown for IBTX sensitivity in dentate gyrus neurons [20,25]. On the basis of widespread putative IBTX-resistant β4 subunit expression, but comparatively rarer examples of IBTX-resistant channels, it has also been proposed that α subunits can be less than fully saturated with β4, and, thus, that IBTX can still block such channels [12]. It is also possible that BK_Ca_ diversity may extend to mixed accessory subunits, such as β2 and β4 heteromultimers.

BK_Ca_ channels are key contributors to neuronal excitability, and it may be predicted that BK_Ca_ opening would cause general hyperpolarization and thereby reduce neuronal hyperexcitability, as proposed for VSN16R effects in spasticity [2]. Such actions may prevent epileptiform discharges and be useful agents for seizure control. For example, BK_Ca_ loss-of-function has been implicated in conditions such as temporal lobe epilepsy [26]. Our data demonstrate that VSN16R increases and IBTX decreases firing frequency in CA1 neurons. Although this may appear counter-intuitive, BK_Ca_ channels have been shown to facilitate high frequency firing of CA1 neurons via effects on rapid spike repolarization and the fast afterhyperpolarization, which ultimately reduce Na^+^ channel inactivation and activate slower, delayed rectifier K^+^ channels [8]. In general, for BK_Ca_ channels it appears that a correct balance of excitability must be maintained, as both loss-of-function and gain-of-function BK_Ca_ mutations have been associated with different forms of epilepsy; correspondingly, pharmacological intervention by BK_Ca_ openers and inhibitors have reported utility in preventing epileptiform discharges in different models (see [27,28,29]. Overall, increasing our pharmacological knowledge of BK_Ca_ channel activators and inhibitors and their role(s) in modulating different forms of epilepsy, and other diseases of hyperexcitability, and of how different BK heteromonomers and/or heteromultimers may be targeted selectively, is a key aim. Here, we provide further evidence that the cannabinoid-like VSN16R compound, which has demonstrated good CNS penetration, may represent a useful starting point for therapeutic development of BK_Ca_ openers.

## Figures and Tables

**Figure 1 pharmaceuticals-12-00104-f001:**
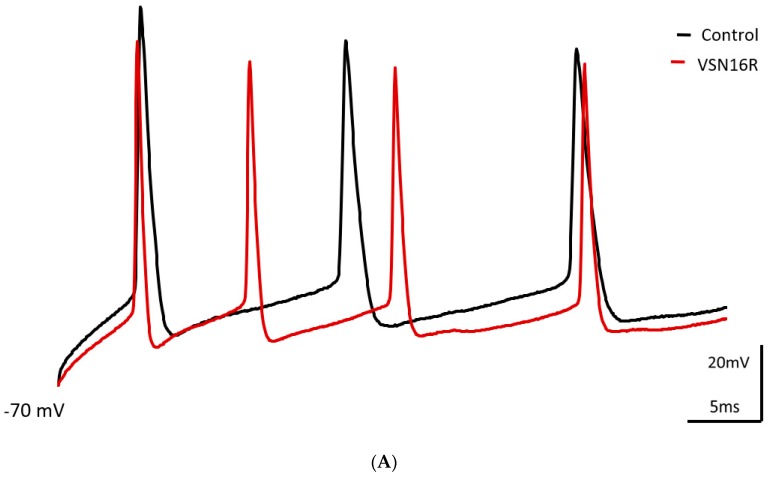
Effect of VSN16R on AP waveform under high frequency stimulus. (**A**) Exemplar trace showing effects of 100 μM VSN16R. (**B**) Effects before (control) and at steady state after application of 100 μM VSN16R on absolute change of first and second fAHP amplitude from resting potential, first and second ISI and first and second AP width. Data from n = 7 cells; * = *P* < 0.05; ** = *P* < 0.01.

**Figure 2 pharmaceuticals-12-00104-f002:**
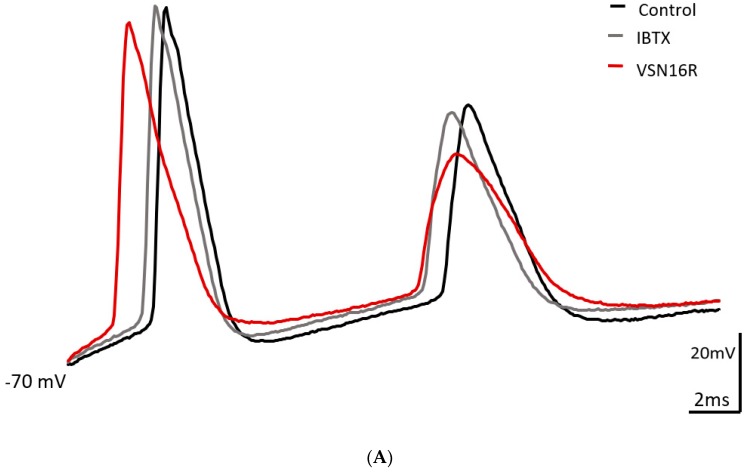
Effects of IBTX and VSN16R on AP waveform under high frequency stimulus. (**A**) Exemplar trace showing effects of 100 nM IBTX and subsequent application of 100 μM VSN16R. (**B**) Effects before (control) and at steady state after application of 100 nM IBTX and 100 μM VSN16R on absolute change of first fAHP amplitude from resting potential, second ISI and first AP width. Data from n = 5 cells; * = *P* < 0.05; ** = *P* < 0.01.

**Figure 3 pharmaceuticals-12-00104-f003:**
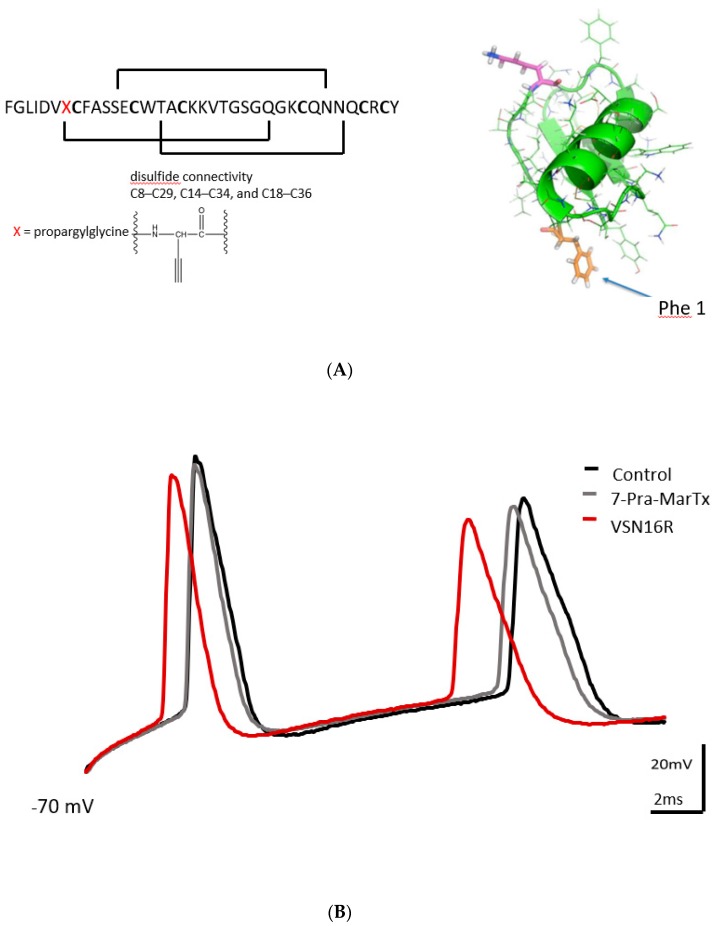
Effects of 7-Pra-MarTx and VSN16R on AP waveform under high and low frequency stimulus. (**A**) Amino acid sequence, connectivity and ribbon diagram showing 3D structure of 7-Pra-MarTx. (**B**) Exemplar trace showing effects of 7-Pra-MarTx and subsequent application of 100 μM VSN16R under high frequency stimulus. (**C**) Effects before (control) and at steady state after application of 100 nM 7-Pra-MarTx and 100 μM VSN16R on absolute change of first fAHP amplitude from resting potential and first ISI under low frequency stimulus. Data from n = 7 cells; * = *P* < 0.05.

**Table 1 pharmaceuticals-12-00104-t001:** Effects of (R,Z)-3-(6-(dimethylamino)-6-oxohex-1-en-1-yl)-N-(1-hydroxypropan-2-yl) benzamide (VSN16R) on action potential (AP) waveform parameters.

AP Waveform Parameter	High Frequency Stimuli		Low Frequency Stimuli	
	Control	VSN16R	Control	VSN16R
**fAHP1 (mV)**	17.0 ± 1.7	14.3 ± 1.6 *	16.5 ± 1.7	17.2 ± 1.4
**fAHP2 (mV)**	21.9 ± 1.7	19.0 ± 1.6 *	20.5 ± 1.6	20.2 ± 2.1
**ISI1 (ms)**	16.3 ± 1.3	12.7 ± 1.2 **	30.4 ± 4.3	26.4 ± 4.4
**ISI2 (ms)**	20.1 ± 0.8	16.9 ± 1.3 *	31.2 ± 2.1	29.2 ± 2.1
**AP1 width (ms)**	4.4 ± 0.4	3.3 ± 0.5 *	4.4 ± 0.3	4.0 ± 0.4
**AP2 width (ms)**	5.5 ± 0.5	4.3 ± 0.8	5.2 ± 0.5	5.0 ± 0.5

Data from n = 7 cells; * = *P* < 0.05; ** = *P* < 0.01.

**Table 2 pharmaceuticals-12-00104-t002:** Effects of iberiotoxin (IBTX) and subsequent application of VSN16R on AP waveform parameters.

AP Waveform Parameter	High Frequency Stimuli			Low Frequency Stimuli		
	Control	IBTX	VSN16R	Control	IBTX	VSN16R
**fAHP1** **(mV)**	11.5 ± 1.8	20.5 ± 4.4 *	16.1 ± 12.4	11.6 ± 2.2	13.6 ± 2.4	14.3 ± 2.5
**fAHP2** **(mV)**	18.7 ± 1.5	22.9 ± 3.0	21.1 ± 2.1	17.6 ± 2.5	22.7 ± 4.3	18.0 ± 2.4
**ISI1** **(ms)**	10.0 ± 0.7	10.9 ± 0.9	10.9 ± 0.9	12.2 ± 1.0	13.2 ± 1.1	12.2 ± 1.1
**ISI2** **(ms)**	12.7 ± 1.0	14.4 ± 1.1 *	14.7 ± 1.3	17.0 ± 1.1	17.9 ± 1.4	17.3 ± 1.8
**AP1 width** **(ms)**	3.6 ± 0.3	4.2 ** ± 0.2	3.7 * ± 0.3	3.8 ± 0.4	4.2 * ± 0.3	4.0 ± 0.3
**AP2 width** **(ms)**	4.7 ± 0.4	5.7 ± 0.5	5.3 ± 0.4	4.9 ± 0.4	5.5 * ± 0.5	5.1 ± 0.4

Data from n = 5 cells; * = *P* < 0.05; ** = *P* < 0.01.

**Table 3 pharmaceuticals-12-00104-t003:** Effects of 7-Pra-MarTx and subsequent application of VSN16R on AP waveform parameters.

AP Waveform Parameter	High Frequency Stimuli			Low Frequency Stimuli		
	Control	7-Pra-MarTx	VSN16R	Control	7-Pra-MarTx	VSN16R
**fAHP1** **(mV)**	10.8 ± 1.8	13.2 ± 1.7	10.0 ± 1.2	10.3 ± 1.8	11.9 ± 2.0	8.3 ± 1.5 **
**fAHP2** **(mV)**	18.4 ± 1.5	18.1 ± 1.9	15.5 ± 1.0*	14.8 ± 2.0	15.4 ± 2.0	12.3 ± 1.7 **
**ISI1** **(ms)**	14.8 ± 1.2	14.1 ± 1.3	12.3 ± 0.8 *	27.9 ± 3.9	26.1 ± 4.3	19.1 ± 2.7 *
**ISI2** **(ms)**	20.5 ± 1.7	17.6 ± 1.3	17.8 ± 1.6	37.3 ± 2.6	40.2 ± 3.2	31.3 ± 3.0 *
**AP1 width** **(ms)**	4.0 ± 0.2	3.7 * ± 0.2	3.3 * ± 0.2	4.1 ± 0.3	3.8 ± 0.3	3.7 ± 0.3
**AP2 width** **(ms)**	5.2 ± 0.6	4.9 * ± 0.5	5.0 ± 0.5	5.3 ± 0.5	4.7 * ± 0.4	5.2 ± 0.5

Data from n = 7 cells; * = *P* < 0.05; ** = *P* < 0.01.

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
