# Peer review of "The Cannabinoid-Like Compound, VSN16R, Acts on Large Conductance, Ca2+-Activated K+ Channels to Modulate Hippocampal CA1 Pyramidal Neuron Firing"

_pharmaceuticals, 2019, doi:10.3390/ph12030104_

Round 1

Reviewer 1 Report

Dear authors, the work is well done, but the main objective of this manuscript is not clear to me. Several natural compounds are known to activate neuronal BK channels. The development of novel BK channel openers for CNS disorders requires experiments performed testing the effects of several hundred molecules in HTS using the automated planar patch. Your compound is not potent in comparison to other known BK channel openers of natural origin or synthetic one. Most of the BK channel openers available failed in the preclinical setting and few failed in clinical trials as antistroke drugs.  

The main objective of this work needs to be clearly defined in the introduction section and the results consequently discussed. 

Author Response

The main objective of this work needs to be clearly defined in the introduction section and the results consequently discussed. 

We appreciate the reviewer's major point. Here, our main objective is to test the cannabinoid-like compound VSN16R action on native BKCa channels and to add to the literature regarding potential mechanisms of action for cannabinoid compounds. In this regard, cannabinoids are often described as having a 'polypharmacology' (and indeed often unspecified mode of action) and our paper adds to the field in this regard. We have now clarified this point in the first paragraph of the Abstract to better describe our research question. We have also added a further sentence in the Discussion such that "This study further confirmed that cannabinoid-like compounds include BKCa channels amongst their repertoire of molecular targets".

Reviewer 2 Report

The study entitled The cannabinoid-like compound, VSN16R, acts on large conductance, 
Ca2+-activated K+ channels to modulate hippocampal CA1 pyramidal neuron firing
, describes the effect of VSN16R (a BKCa channel opener) on the CA1 pyramidal neurons in hippocampal slices. By means of patch-clamp recordings, the article analyses the changes induced by VSN16R on the following electrophysiological parameters∶ the after-hyperpolarization amplitude, the inter spike interval and the action potential width. The study is well designed and experiments are sound. 

Authors produced evidence that VSN16R may differentially target BKCa channel subtypes and taking into account the CNS penetration of the compound they proposed the use of VSN16R as an useful starting point for therapeutic development of BKCa channels openers.

I recommend the publication on the paper in the present form.

Author Response

We thank the reviewer for positive comments.